environmental science/geology

discontinuous surface deformation, thick and hard conglomerate, surface cracking, thin surface soil, bed separation backfill

**Authors for correspondence:**
Shaojie Chen
e-mail: csjwyb@163.com
Bo Li
e-mail: 1114654624@qq.com

# Bed separation backfill to reduce surface cracking due to mining under thick and hard conglomerate: a case study

Dawei Yin[1,2], Shaojie Chen[1], Bo Li[1] and Weijia Guo[1]

[1]State Key Laboratory of Mine Disaster Prevention and Control, and [2]Shandong Key Laboratory of Civil Engineering Disaster Prevention and Mitigation, Shandong University of Science and Technology, Qingdao 266590, People's Republic of China

DY, 0000-0002-8846-2001; SC, 0000-0003-1377-0808;
BL, 0000-0002-4094-3607

After coal mining, the surface above a goaf may experience the discontinuous deformation under some special geological and mining conditions, such as surface cracking, surface step subsidence and collapse pits. Discontinuous deformation seriously threatens the safety of surface buildings and infrastructures. In this paper, the mechanism of discontinuous surface deformation and surface cracking due to coal mining under thick and hard conglomerate in the Huafeng coal mine was studied using a simulation test on similar materials. Bed separation backfill was then proposed to control surface cracking and to protect the Luli bridge. Because of lithological differences between the conglomerate and relatively weak red strata (beneath the conglomerate), the bed separation occurred between them with the advancement of the working face. When the bed separation span exceeded its breaking span, the conglomerate fractured, causing surface cracking of the downhill area and seriously damaging the stability of the Luli bridge. Three drilling holes were arranged along the strikes of the 1412 and 1613 working faces and nearly 387 000 m$^3$ of backfill materials (water, fly ash and gangue powder) were injected into the bed separation space to reduce or prevent fracturing of the conglomerate. The compacted backfill body supported the conglomerate and reduced the subsidence of the basin and surface 'rebound' deformation at the edge of the subsidence basin. Clay in the red strata expanded upon contact with water, and this further backfilled the bed separation zone and supported the conglomerate. The upper and lower structures and foundation of the bridge were

reinforced using various methods. It was shown that bed separation backfill effectively controlled conglomerate movement and protected the bridge with a maximum subsidence of 251 mm. No obvious surface cracks were observed near the Luli bridge.

# 1. Introduction

After coal mining, the surface above a goaf generally experiences continuous deformation, such as continuous subsidence, horizontal movement and tilt [1–6]. Under some special geological and mining conditions, the surface may have discontinuous deformation, such as surface cracking, surface step subsidence and collapse pits [7–10]. Continuous and discontinuous surface deformations (especially discontinuous deformation) can damage surface buildings and infrastructure. Therefore, to protect surface buildings and infrastructures, it is important to study the mechanisms and controlling methods of discontinuous surface deformation.

Numerous studies have been conducted by many domestic and foreign scholars regarding the mechanisms and controlling methods of discontinuous surface deformation. In Korea, Jung *et al.* developed a quick, simple and quantitative method for estimating subsidence susceptibility and suggested that surface tension cracking was a kind of discontinuous subsidence [11]. Donnelly and Reddish found that the surface step subsidence that affected surface topography may be centred over contrasting lithological contacts such as the bedding planes, joints, unconformities and outcropping of fold axes [12]. Kotyrba & Kortas found that the formation of a horizontal tension zone in a rock mass and convex flexure of a subsidence basin surface caused the discontinuous surface deformation in the form of fissures and step subsidence [13]. Dai and Zuo *et al.* proposed that the occurrence and structural form of the coal and rock layers were internal conditions for discontinuous surface deformation, and that mining-induced influences were external conditions [7,14]. Also, a high-angle fault, steeply inclined coal seam and weak intercalated layers in the coal and rock layers were potential conditions for discontinuous surface deformation. When a steeply inclined coal seam was mined, the uneven mining thickness may have caused inconsistency in failure heights in the overlying strata, and a funnel-shaped collapse pit may appear at the surface [15]. Kowalski *et al.* summarized and analysed the discontinuities in surface deformation characteristics in the Upper Silesian Coal Basin, including the crevices and steps [16]. Li *et al.* found that both the fault structures and alteration contact zones were the main reasons for surface cracking in Nickel Mine [17]. They used the backfill mining to control surface cracking. Under the influence of fault, the subsidence showed an apparent discontinuity in the form of step subsidence near the fault [18]. Guo *et al.* suggested that the surface cracking and step subsidence in Zhaizhen coal mine were induced by fault activation, which was caused by repeated mining [19]. Gangue backfill mining was used to control surface cracking and step subsidence in Zhaizhen coal mine.

According to the above investigations, the discontinuous surface deformation occurs under some special geological and mining conditions, such as coal mining near faults, mining steeply inclined coal seams, etc. The overlying strata in Huafeng coal mine located in Shandong Province of China are composed of a Tertiary conglomerate (400–800 m) and a thin surface soil (0–8 m). There is no special geological structure. Some discontinuous surface deformations develop in this mining area after coal mining is conducted here. Surface cracks are present on the downhill surface (northern direction of the working face). Additionally, there is one bridge crossing the northern Huafeng coal mine. Discontinuous surface deformation has a critical effect on the bridge stability and safety. Once one part of a bridge (such as a beam or pier) moves or tilts, the whole bridge will lose stability. In this paper, the discontinuous surface deformation mechanism of the Huafeng coal mine was analysed using a simulation test with similar materials. Also, bed separation backfill was proposed and used to control surface cracking and to protect the bridge. The mechanism by which bed separation backfill controlled surface cracking was discussed.

# 2. Engineering background

## 2.1. Geological and mining conditions in Huafeng coal mine

The Huafeng coal mine is located in Taian of Shandong Province of China. A comprehensive stratigraphic column of the Huafeng coal mine is shown in figure 1. The mining depth of the Huafeng

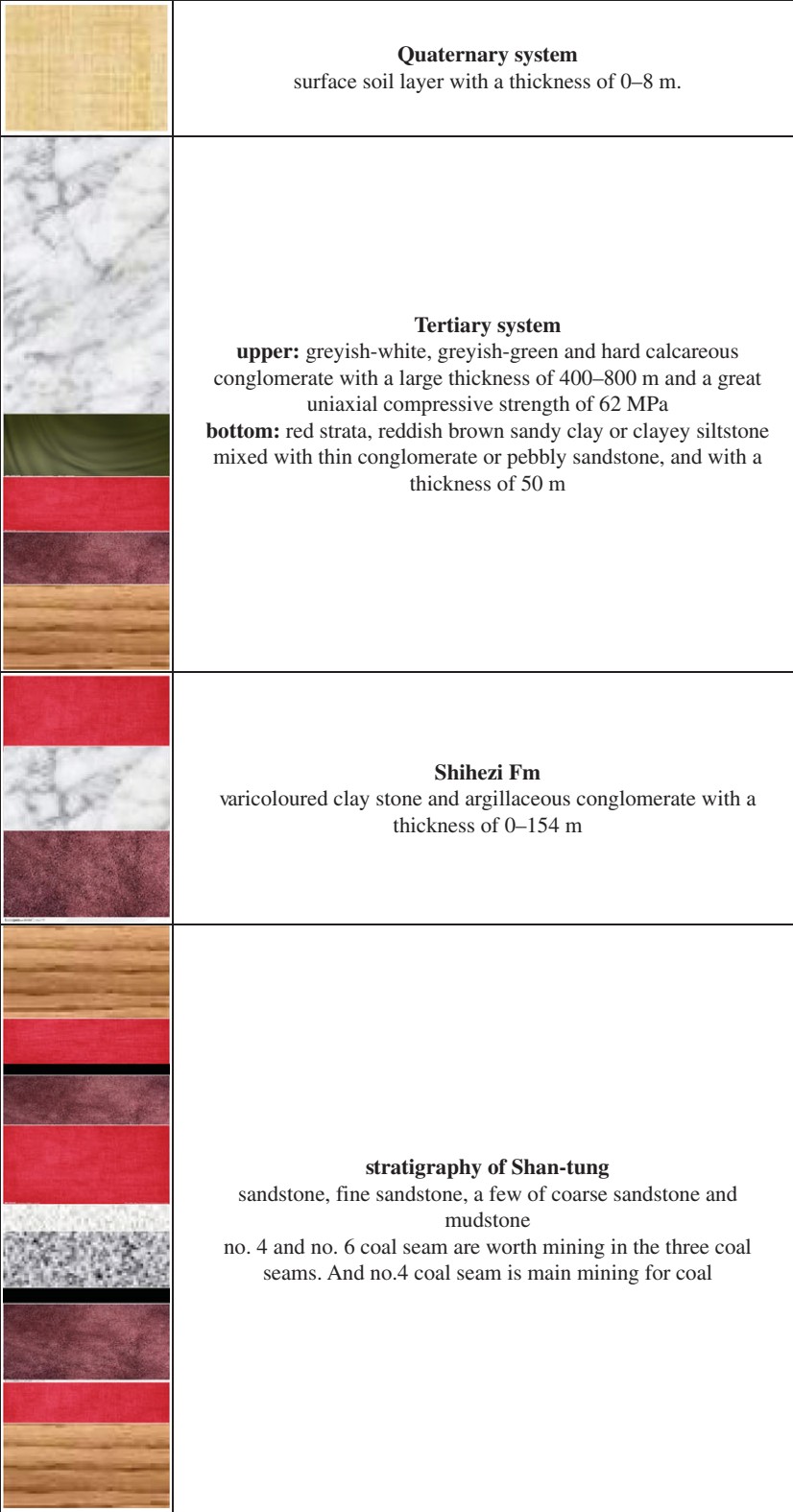

**Figure 1.** Comprehensive stratigraphic column of the Huafeng coal mine.

coal mine is approximately −1300 m (below the surface). The main coal seam is the no. 4 coal seam, which has an average thickness of 6.4 m. The coal seam is overlain by the Tertiary conglomerate with a thickness of 400–800 m and Quaternary soil with a thin thickness of 0–8 m. The average uniaxial compressive strength of the conglomerate is 62 MPa. Relatively weak 'red strata' have an average

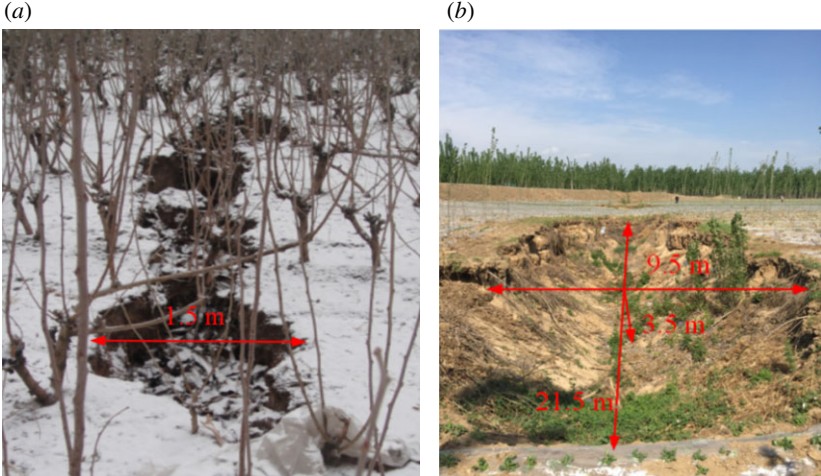

**Figure 2.** Typical discontinuous surface deformations above the Huafeng coal mine. (*a*) Surface cracking [20], (*b*) Sunken pit induced by surface cracking.

thickness of about 50 m, which consists of the reddish brown sandy clay or clayey siltstone mixed with thin conglomerate or pebbly sandstone, and are located beneath the conglomerate.

## 2.2. Characteristics of discontinuous surface deformation

The typical discontinuous surface deformation found in the Huafeng coal mine is surface cracking (figure 2). The surface cracks have the following characteristics:

(1) Surface cracks appear on the downhill area surface (northern direction of the working face) when the working face is advanced 400–600 m. Cracks widen to 1.5–2.5 m after coal mining. If the working face continues to progress down-dip, primary cracks experience compression and then slowly close. Otherwise, surface soil collapses into the surface cracks to form a sunken pit (figure 2*b*).
(2) A continuous surface crack runs parallel to the strike of the coal seam, and the development direction range is about 100°–105° along the trend of the fracture. A new crack appears 60–80 m away from the initial surface crack.
(3) Larger cracks are located in areas of large horizontal tensile deformation, and the tensile deformation values of the area where surface cracking occurs generally exceed 2.8 mm m$^{-1}$. The angle between the crack and the lower roadway that extends to the working face is 64°–68°.

## 2.3. Surface cracking near the Luli bridge

The Luli bridge is located in the northern mining area (figures 3 and 4). The length of the bridge is 602 m and it has 32 spans that are 13 m apart. The main bridge length is 429 m and the approach bridge length is 173 m. Additionally, the bridge width is 8 m, and there are 64 bridge pillars. The Luli bridge is a simple concrete-supported beam bridge, which was built by the Tai'an government in October 1990. A total of 8.82 million tons of coal can be extracted from the protective coal pillar under the bridge.

There are currently two surface cracks that can be observed near the Luli bridge, and these were induced by mining of the 1411 and 1612 working faces (figure 3). According to surface cracking characteristics, the surface cracks caused by the mining of 1412 and 1613 working faces were estimated, respectively, as shown in figure 3. These surface cracks will pass beneath the Luli bridge and affect its stability and threaten the safety of pedestrians and vehicles. Generally, when a surface crack propagates beneath the bridge, the bridge foundation may be displaced, generating additional stress in the bridge. When the additional stress exceeds the tensile strength of the concrete structure, the bridge will fracture. Generally, the top or bottom of the bridge deck pavement is cracked. Moreover, the large surface cracks may cause a significant vertical displacement between two bridge plates. If the vertical distance is larger than the capping beam width, the bridge plates will fall, and this is a serious threat to the safety of pedestrians and vehicles. In addition, bridge piers and

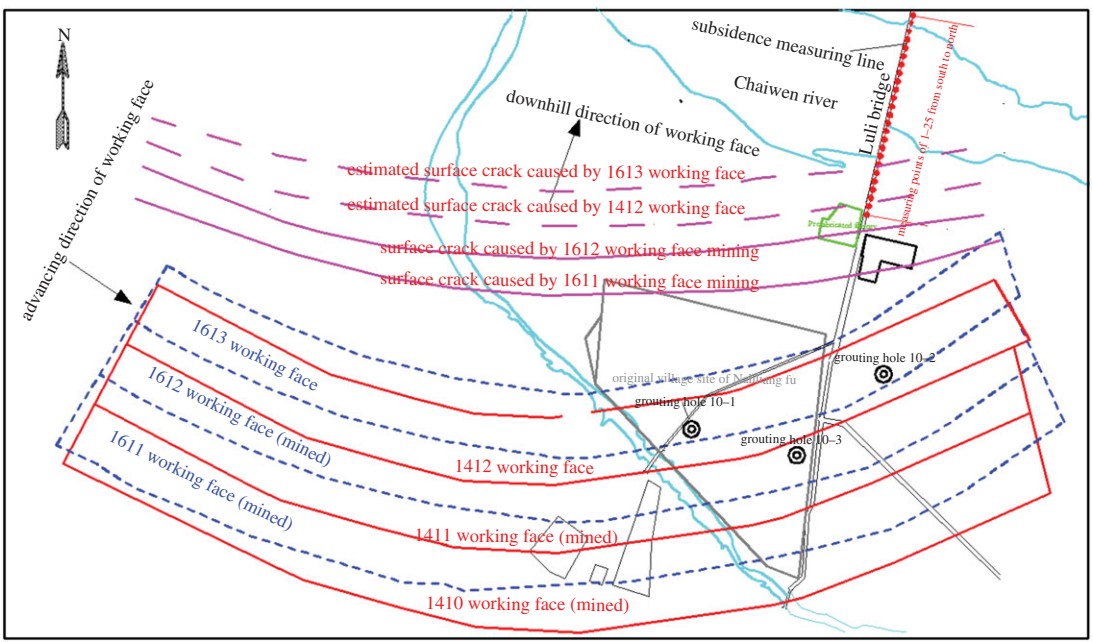

**Figure 3.** Schematic diagram showing the relationship between the Luli bridge and working faces in plane view.

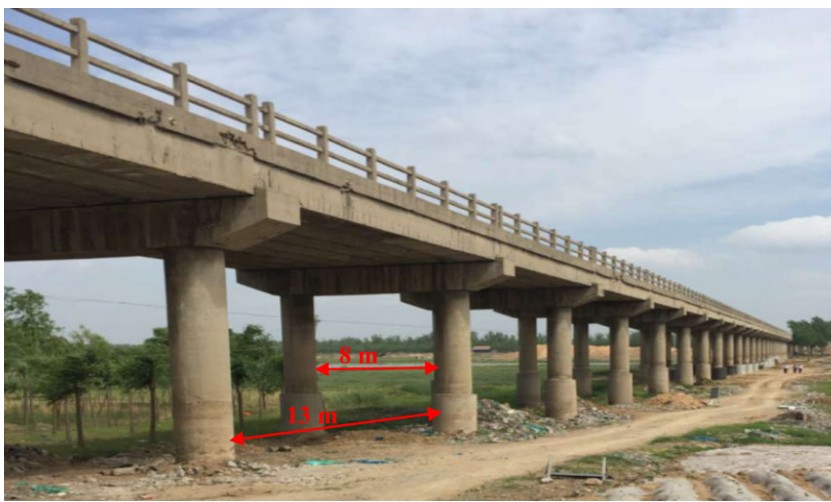

**Figure 4.** Photograph of the Luli bridge.

foundations are subjected to shear stress induced by rock and soil deformation, which seriously damages the piers and foundations of the bridge.

To protect the Luli bridge, we should first understand the discontinuous surface deformation mechanism that is caused by coal mining. An experiment using similar materials was conducted to analyse the discontinuous surface deformation mechanism, and discussion of this experiment and the analysis follows.

# 3. Simulation test using similar materials to study the discontinuous surface deformation mechanism

## 3.1. Model frame and similarity coefficient

Based on the geological conditions of the 1410 working face in the Huafeng coal mine, a simulation test using similar materials was conducted to study the mechanism of discontinuous surface deformation.

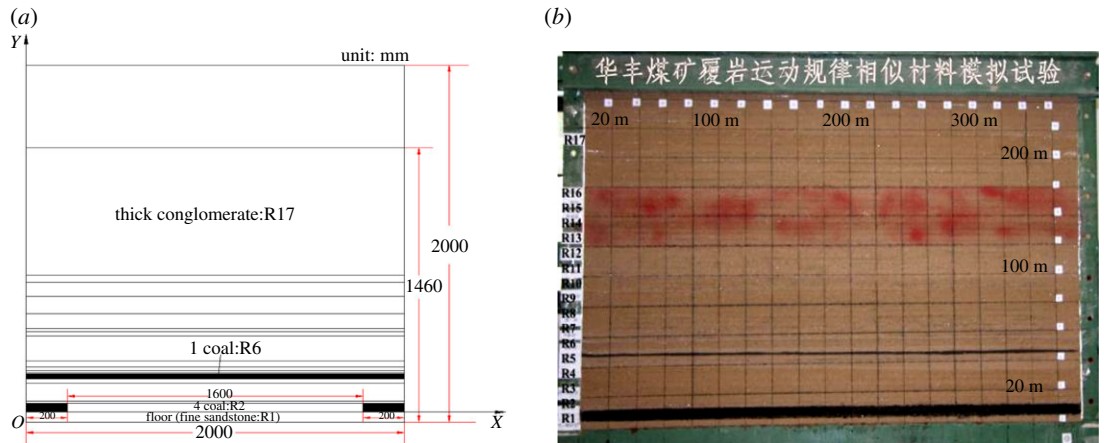

**Figure 5.** Experimental and physical models for the simulation test using similar materials. (a) Schematic diagram of the experimental model, (b) image of the physical model.

The size of the two-dimensional model frame was 2000 mm long × 2200 mm wide × 2000 mm high. Experimental and physical models of the similar materials are shown in figure 5.

The lengths of 1410 working face and left coal pillars were 320 m and 40 m, respectively. Considering the experimental device, the constant of geometric similarity $C_L$ was taken as

$$C_L = \frac{X_m}{X_P} = \frac{Y_m}{Y_P} = \frac{1}{200}, \tag{3.1}$$

where $X_m$ and $Y_m$ are the lengths of the physical model in two mutually perpendicular directions; $X_P$, $Y_P$ are the lengths of two mutually perpendicular directions of the prototype.

Thus, in the model, the lengths of the mined coal seam and coal pillars were 1600 mm and 200 mm, respectively.

According to the properties of the selected simulation materials and their ratios, the constant of bulk density similarity $C_\gamma$ was taken as

$$C_\gamma = \frac{\gamma_{mi}}{\gamma_{pi}} = \frac{1}{1.5}, \tag{3.2}$$

where $\gamma_{mi}$ is the specific gravity of the $i$-th stratum in the prototype, and $\gamma_{pi}$ is the corresponding specific gravity of the strata in the model.

According to equations (3.1) and (3.2), the constant of stress similarity $C_{\sigma e}$ and constant of time similarity $C_t$ should satisfy the following equations:

$$C_{\sigma e} = C_L C_\gamma = \frac{1}{300} \tag{3.3}$$

and

$$C_t = \frac{T_m}{T_P} = \sqrt{C_L} \approx \frac{1}{14.14}, \tag{3.4}$$

where $T_m$ and $T_p$ are the mining time of the model and field, respectively. For the convenience of research, $C_t$ was taken as 1/14, which means that a modelling time of 1 h represents for an actual mining time of 14 h.

Table 1 gives the parameters of the physical model [21]. It is important to note that the conglomerate is significantly stronger than adjacent rock units. Thus, the ratios of the lime were enhanced to simulate the behaviour of the conglomerate. The weak red strata corresponded to R13–R16 layers in the model and were coloured red (figure 5b). The main coal seam was the no. 4 coal seam.

**Table 1.** Proportion, materials and layout order of the model strata [21].

| stratum | lithology | actual thickness (m) | thickness in simulated model (cm) | layered number | layered thickness (cm) | bulk density of similar materials (g cm$^{-3}$) | material consumption (kg) | | | | uniaxial compressive strength of the prototype (MPa) | uniaxial compressive strength of the model (KPa) |
| --- | --- | --- | --- | --- | --- | --- | --- | --- | --- | --- | --- | --- |
| | | | | | | | sand | lime | gypsum | water | | |
| R17 | thick conglomerate | 144 | 72 | 18 | 4 | 1.45 | 28.44 | 1.78 | 1.78 | 2.90 | 62.0 | 105.31 |
| R16 | clayey siltstone | 8.35 | 4.2 | 2 | 2.1 | 1.5 | 15.45 | 1.16 | 0.77 | 1.58 | 32.2 | 54.69 |
| R15 | mudstone | 15.6 | 7.8 | 3 | 2.6 | 1.5 | 19.13 | 1.67 | 0.72 | 1.96 | 25.6 | 43.48 |
| R14 | clayey siltstone | 20.0 | 10 | 5 | 2 | 1.5 | 37.01 | 2.78 | 1.84 | 3.78 | 32.2 | 54.69 |
| R13 | mudstone | 16.55 | 8.3 | 5 | 1.66 | 1.5 | 20.29 | 1.78 | 0.77 | 2.09 | 25.6 | 43.48 |
| R12 | medium grain sandstone | 3.85 | 1.9 | 2 | 0.95 | 1.6 | 7.45 | 0.47 | 0.47 | 0.76 | 36.6 | 62.17 |
| R11 | siltstone | 4.6 | 2.3 | 2 | 1.15 | 1.5 | 8.46 | 0.63 | 0.42 | 0.87 | 43.4 | 73.72 |
| R10 | medium grain sandstone | 28.85 | 14.4 | 8 | 1.8 | 1.6 | 55.83 | 3.52 | 3.52 | 5.70 | 36.6 | 62.17 |
| R9 | mudstone | 6.6 | 3.3 | 2 | 1.65 | 1.5 | 8.09 | 0.71 | 0.30 | 0.83 | 25.6 | 43.48 |
| R8 | medium grain sandstone | 3.95 | 2 | 2 | 1.0 | 1.6 | 7.85 | 0.49 | 0.49 | 0.80 | 36.6 | 62.17 |
| R7 | siltstone | 1.17 | 1.5 | 2 | 0.75 | 1.5 | 5.61 | 0.42 | 0.28 | 0.58 | 43.4 | 73.72 |
| R6 | coal seam 6 | 1.20 | 0.6 | 1 | 0.6 | 1.35 | 3.98 | 0.34 | 0.15 | 0.41 | 8.42 | 14.30 |
| R5 | siltstone | 5.29 | 2.6 | 2 | 1.3 | 1.5 | 9.73 | 0.72 | 0.48 | 1.00 | 43.4 | 73.72 |
| R4 | sandy mudstone | 5.25 | 2.6 | 2 | 1.3 | 1.5 | 9.56 | 0.84 | 0.36 | 0.98 | 23.2 | 39.41 |
| R3 | mid-fine grained sandstone | 20 | 10 | 5 | 2 | 1.6 | 15.69 | 0.98 | 0.98 | 1.61 | 48.48 | 82.34 |
| R2 | siltstone | 2.6 | 1.3 | 1 | 1.3 | 1.5 | 4.78 | 0.36 | 0.24 | 0.49 | 43.4 | 73.72 |
| R1 | coal seam 4 | 6.41 | 3.2 | 1 | 3.2 | 1.35 | 21.19 | 1.85 | 0.79 | 2.17 | 10.4 | 17.66 |
| floor | fine sandstone | 10 | 5 | 2 | 2.5 | 1.5 | 19.55 | 1.23 | 1.23 | 2.01 | 51.2 | 86.96 |

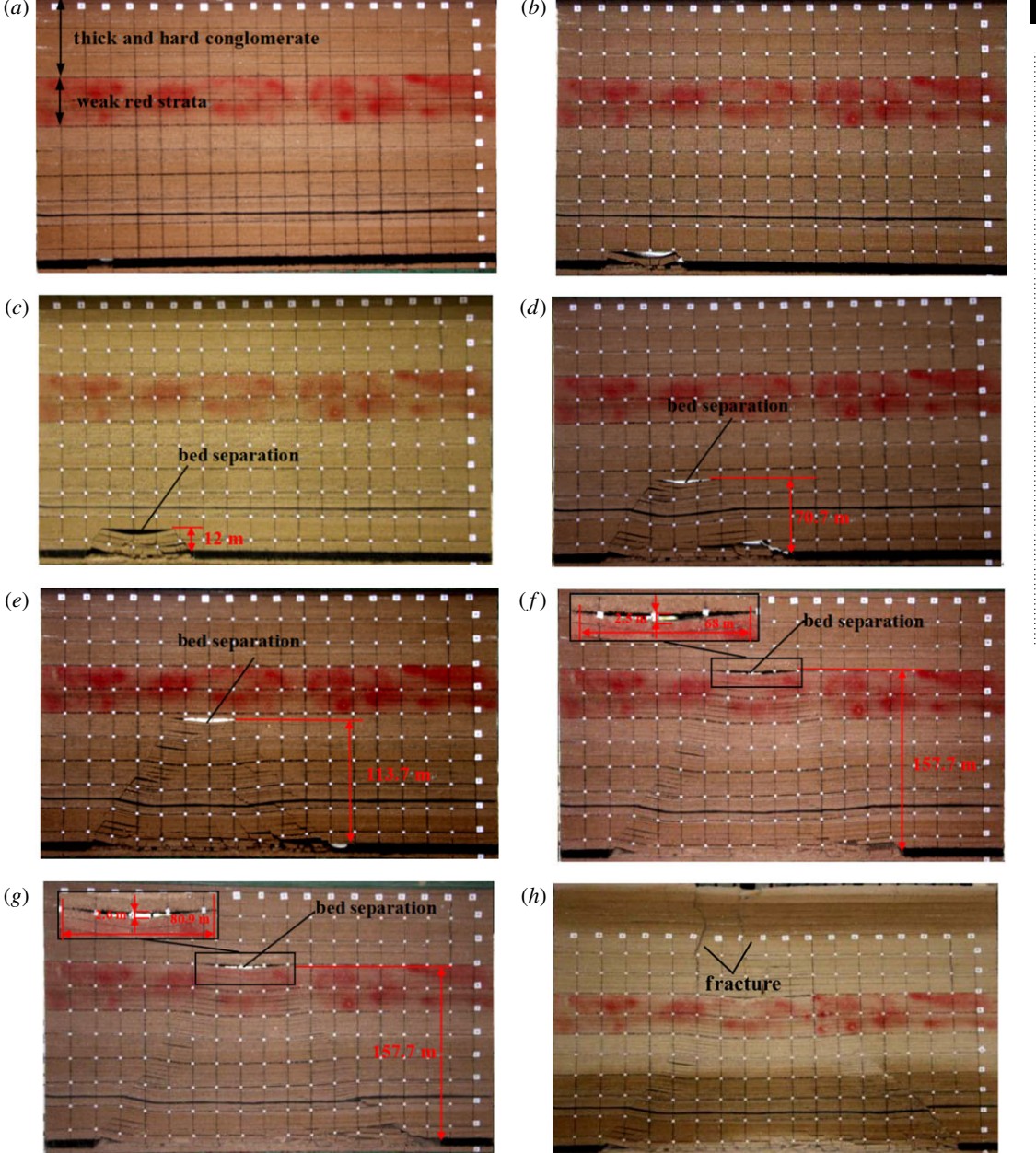

**Figure 6.** Photographs of the physical model showing the movement of the overlying strata. (*a*) open-off cut, (*b*) advance distance of 40 m, (*c*) advance distance of 70 m, (*d*) advance distance of 150 m, (*e*) advance distance of 220 m, (*f*) advance distance of 250 m, (*g*) advance distance of 290 m, (*h*) advance distance of 320 m.

## 3.2. Test results and analysis

### 3.2.1. Movement characteristics of overlying strata

The movement of overlying strata during the simulated mining excavation is shown in figure 6. When the coal seam was mined, the range of influence of coal mining increased from the roof to overlying strata. Characteristics of the movement of overlying strata are as follows:

(a) When the working face advanced to 40 m from the opening cut (figure 6*b*), the roof immediately collapsed, illustrating that the first weighting step distance was 40 m in the experiment. This is consistent with the first field weighting step distance of 43 m [20].

(b) When the working face advanced to 70 m (figure 6*c*), the roof above the goaf collapsed with a caving height of 12 m. Because of the differences in lithology, the bed separation will occur between adjacent rock units.

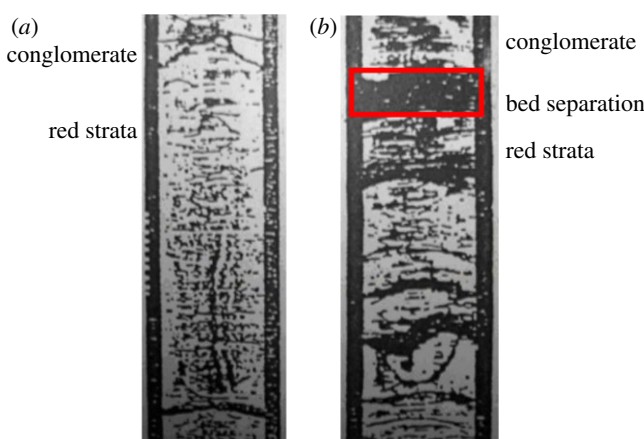

**Figure 7.** Ultrasonic images of borehole pre- and post-mining [20]. (*a*) Before coal mining, (*b*) after coal mining.

(c) As the working face advanced further, the range of influence of coal mining on overlying strata developed upward. Bed separation at the bottom of the model was compacted, and a new bed separation occurred above the first bed separation. When the working face was advanced to 250 m (figure 6*f*), the weak red strata sank, and strata below them were further compacted. Moreover, bed separation with a height of 2.5 m and a length of 68 m occurred between the conglomerate and weak red strata. The conglomerate formed a suspended roof supported by the lower rock mass at both ends. As the working face continued to advance to 290 m (figure 6*g*), the bed separation developed and expanded to a height of 2.6 m and a length of 80.9 m. The conglomerate became unstable and began to bend. When the working face advanced to 320 m (figure 6*h*), the conglomerate fractured. Factures developed up to the upper boundary of the model and formed surface cracks.

Additionally, the original geophysical logs for the exploration of well bores were compared with new logs of the boreholes to monitor the development of bed separation (figure 7) [15]. Before coal mining, the overlying strata above the coal seam maintained coherency. However, after coal mining, there was a significant bed separation between the conglomerate and red strata. When the bed separation span was larger than the broken step distance of the conglomerate, the conglomerate will fracture, resulting in surface cracking.

### 3.2.2. Discussion of movement characteristics of overlying strata

A motion model of the movement of overlying strata because of mining under the conglomerate was established on the basis of experimental results and is shown in figure 8.

Because of the special geological and mining conditions in the Huafeng coal mine, a composite structure of 'upper-hard and lower-soft strata' was formed, and this was composed of the conglomerate and weak red strata. After coal mining, the red strata ('lower-soft' strata) experienced significant subsidence, whereas the conglomerate ('upper-hard' strata) experienced less subsidence at a relatively slow rate [18]. This phenomenon was called non-simultaneous subsidence of overlying strata [1]. Bed separation occurred easily between the conglomerate and red strata, and a suspended roof formed in the conglomerate (figure 8). The peak pressure was transferred to the deepest extent of the roof, forming two stress concentration zones. As the suspended roof bent and sank slowly under its own weight, the conglomerate will exert a 'leverage' effect on the surrounding rock masses with regions A and B as the pivot points (figure 8). Correspondingly, the surface at the edge of subsidence basin showed the 'rebound', which can absorb some deformation from the conglomerate. However, the bed separation expanded as the working face advanced, which provided a huge space for the conglomerate to move.

According to equation (3.5), when the bed separation span exceeded its broken-step distance, the conglomerate fractured and subsided. Now the deadweight of conglomerate above the bed separation was larger than its internal tensile stress. Thin surface soil above the conglomerate cannot

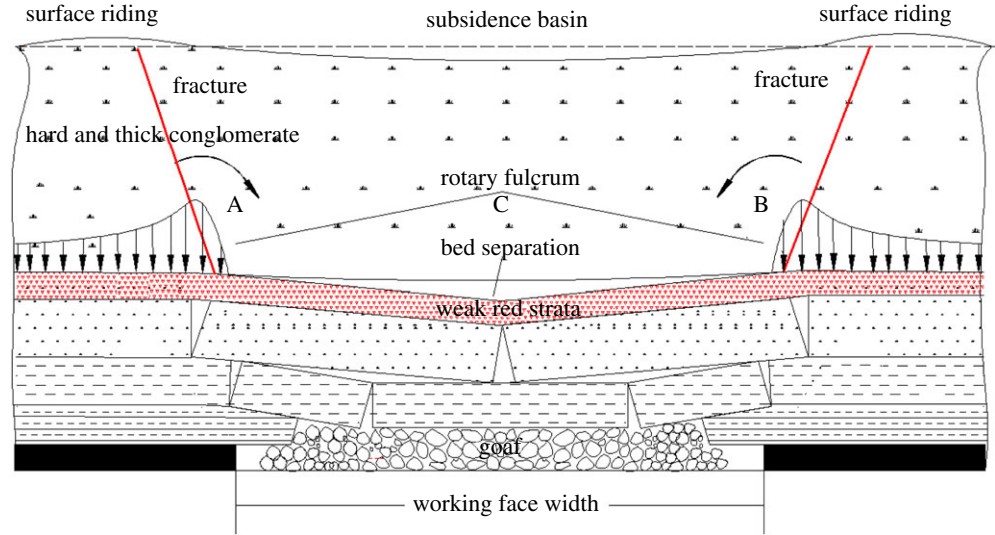

**Figure 8.** Schematic diagram of the motion model for overlying strata movement in the Huafeng coal mine. A and B mark stress concentration zones.

absorb large amounts of deformation. Therefore, a surface crack appeared on the surface area with 'rebound'.

$$\gamma h l > \delta, \tag{3.5}$$

where $\gamma$ and $h$ are the bulk density and height of the conglomerate, respectively; $l$ is the mining width of the working face and $\delta$ is the tensile stress in the conglomerate.

From the above analysis, it is proposed that an effective means for controlling discontinuous surface deformation and protecting the bridge would be to eliminate the space into which the conglomerate could move. Thus, bed separation backfill was proposed and discussion of the analysis follows.

# 4. Bed separation backfill technology and its controlling mechanism on discontinuous surface deformation

## 4.1. Bed separation backfill technology

Bed separation backfill is when backfill materials (water, fly ash and gangue powder) are injected into the bed separation zone using a high-pressure grouting pump (figure 9). This backfill fills the bed separation zone and limits fracturing of the conglomerate, reducing the surface subsidence, subsidence velocity and subsidence range [20,22,23].

To protect the Luli bridge and to control surface deformation, three holes (which were designated 10–1, 10–2 and 10–3) were drilled along the strike of the 1412 and 1613 working faces (figure 3). Drilling along the dip direction was arranged in the position with the maximum prediction of separation development height. Map distances from the drill holes to the bridge and working faces are given in table 2.

A mixture of water, fly ash and waste powder was selected as the backfill material. The water–solid ratio was 1.5. Aggregates were 40% fly ash and 60% gangue powder. A total volume of about 387 000 m$^3$ of grout was injected into the bed separation to protect the Luli bridge and to control surface cracking.

## 4.2. Controlling mechanism of bed separation backfill on discontinuous surface deformation

The control mechanism for how bed separation backfilling affects surface discontinuous deformation has the following aspects:

(1) Backfill materials were injected into the bed separation zone and occupied the bed separation space to limit the movement of the conglomerate, preventing fracturing and subsidence of the conglomerate.

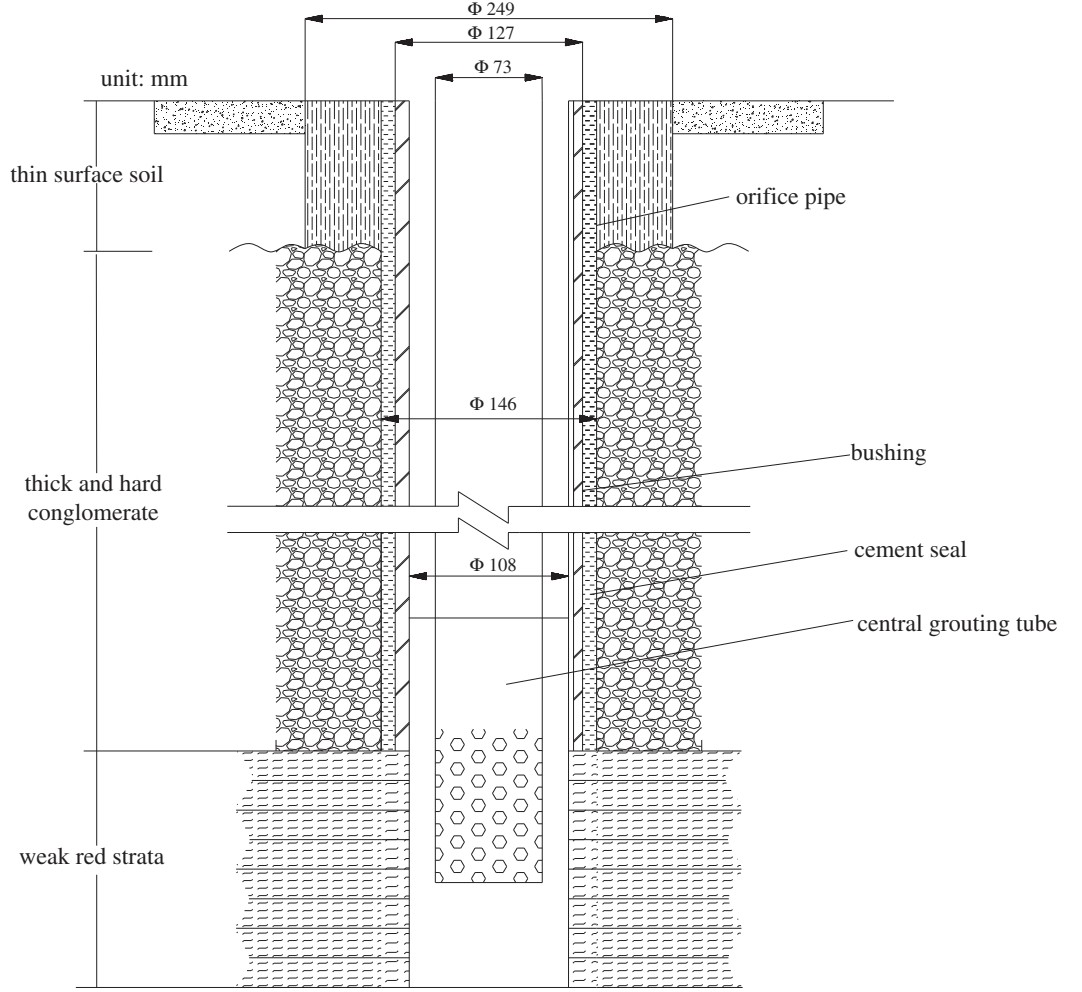

**Figure 9.** Schematic diagram of the drill hole structure used for bed separation backfill.

**Table 2.** Plane map distances from drilling holes to bridges and working faces.

| drill hole | to the Luli bridge | to the trend line of the Luli bridge | to the lower roadway of 1612 working face | to the lower roadway of 1411 working face |
|---|---|---|---|---|
| 10-1 | 510 m | 260 m | 34 m | 130 m |
| 10-2 | 470 m | 10 m | 32 m | 127 m |
| 10-3 | 630 m | 140 m | 80 m | 40 m |

(2) After the backfill materials were consolidated, the backfill body in the bed separation zone supported the conglomerate and reduced the surface subsidence. The backfill body transferred parts of the self-weight stress of the conglomerate. Therefore, the degrees of stress concentration at areas A and B were reduced (figure 10). The 'leverage' effect of the subsiding conglomerate on the surrounding rock masses decreased. Correspondingly, the 'rebound' or 'rebound trend' of surface at the edge of the subsidence basin was reduced or prevented. Additionally, delamination of lower strata was compacted.

(3) Clay in red strata disintegrated and expanded when subjected to water. Bed separation zones were further filled with the expanding clay, and this also limited the movement in the conglomerate.

## 4.3. Other protection measures for the bridge

There were several steps involved in reinforcement measures to protect the Luli bridge. The previous bridge deck pavement was cleaned up and a new concrete deck was paved on the bridge (figure 11a).

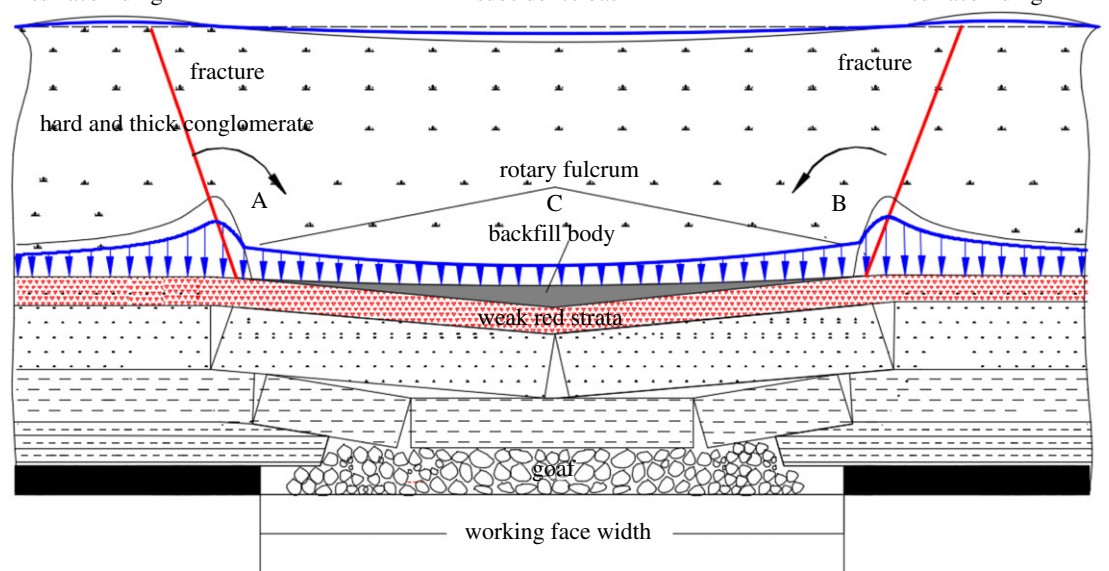

**Figure 10.** Schematic diagram of control mechanism for how bed separation backfilling affects surface discontinuous surface deformation.

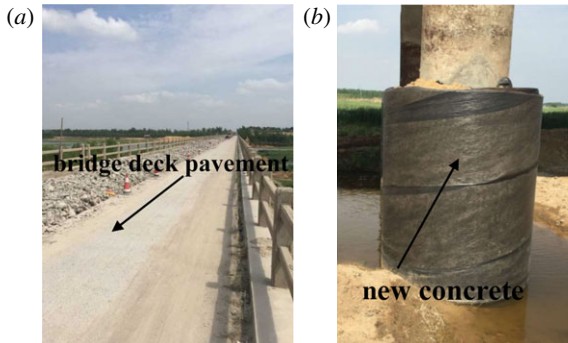

**Figure 11.** Expand section for lower structure reinforcement. (*a*) Pave deck pavement, (*b*) bridge pier.

The influences of mining and the formation of surface cracks are gradual. Lower structural reinforcement can prolong the use of the bridge. Therefore, the cross-section of the bridge pier was widened by pouring concrete along its outer side, as shown in figure 11*b*.

# 5. Observation of surface subsidence along the Luli bridge

A subsidence measuring line with 25 measuring points was arranged along the Luli bridge (figure 3). Figure 12 shows the bridge subsidence after the 1412 and 1613 working faces were mined and bed separation backfilling was completed. The subsidence in the southern area of Luli bridge was larger than that in its northern area. The maximum subsidence of Luli bridge was 251 mm located at the southern end. No obvious surface cracks were observed near the Luli bridge, and this indicates that the bridge has been well protected.

# 6. Conclusion

From the above analysis, the following conclusions can be drawn:

(1) The bed separation occurred easily between the conglomerate and weak red strata as the working face advanced in Huafeng coal mine. When the bed separation span was larger than the broken-step distance of the conglomerate, the conglomerate fractured and caused surface cracking.

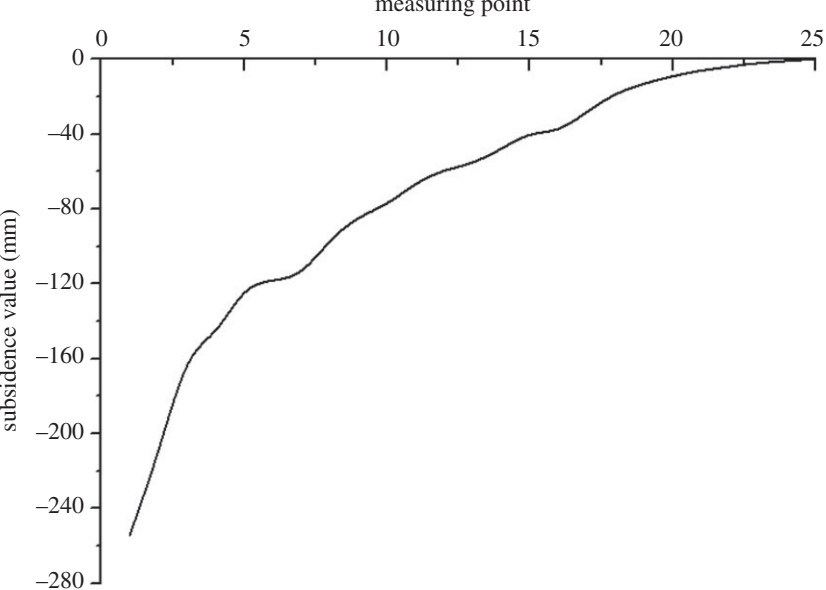

**Figure 12.** Graph of subsidence values of the Luli bridge.

(2) Bed separation backfill reduced the discontinuous surface deformation by occupying bed separation space to reduce or prevent the fracturing of the conglomerate. The compacted backfill body supported the conglomerate, and reduced surface subsidence and surface rebound at the edge of the subsidence basin. Clay in the red strata disintegrated when it was exposed to water, which further backfilled the bed separation and supported the conglomerate.

(3) Three drill holes were arranged along the strike of the 1412 and 1613 working faces. Approximately 387 000 $m^3$ of backfill materials, which mainly consisted of water, fly ash and gangue powder, were injected into the bed separation zone. Additionally, the upper structure, lower structure and foundation of the bridge were reinforced using a variety of methods. It was shown that bed separation backfill effectively controlled the conglomerate movement and protected the bridge with a maximum subsidence of 251 mm. There were no obvious surface cracks observed near the Luli bridge.

Data accessibility. Our data are from the laboratory test and field measurement. The data of similar material simulation test are described in detail in the test results and analysis section (§4.1) of this manuscript. And the data of Luli bridge subsidence are described in detail in the observation surface subsidence along the Luli bridge (§5) of this manuscript.

Authors' contributions. S.C. and D.Y. participated in the test design and data analysis and drafted the manuscript; B.L. and W.G. conducted the tests and analysed the filed measurement data. All authors gave final approval for publication.

Competing interests. The authors declare no competing financial interests.

Funding. S.C. was supported by National Key R&D Programme (no. 2018YFC0604704), Taishan Scholars Project, Taishan Scholar Talent Team Support Plan for Advantaged & Unique Discipline Areas, SDUST Research Fund, National Natural Science Foundation of China (nos. 51474134, 51774194, 51874189), Shandong Provincial Natural Science Fund for Distinguished Young Scholars (no. JQ201612), Shandong Provincial Key Research and Development Plane (no. 2017GSF17112). D.Y. was supported by National Natural Science Foundation of China (no. 51874189) and Opening Foundation of Shandong Key Laboratory of Civil Engineering Disaster Prevention and Mitigation (CDPM2019ZR04).

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
