## [Reviewer comments · Royal Society Open Science]

Review History

RSOS-190880.R0 (Original submission)

Review form: Reviewer 1

Is the manuscript scientifically sound in its present form?

Yes

Are the interpretations and conclusions justified by the results?

Yes

Is the language acceptable?

Yes

Is it clear how to access all supporting data?

Yes

Do you have any ethical concerns with this paper?

No

Have you any concerns about statistical analyses in this paper?

No

Recommendation?

Accept as is

Comments to the Author(s)

The article "Bed separation backfill to reduce surface cracking due to mining under thick and hard conglomerate - a case study" is written in a clear and understandable way.

In this paper, the discontinuous surface deformation mechanism in the Huafeng coal mine was analyzed through a similar material simulation test. Based on these, the bed separation backfill was proposed and adopted to control surface deformation and to protect the Luli bridge. The mechanism of surface cracking controlled by bed separation backfill was discussed.

Formal remarks:

- the summary reflects the essence of the study,
- the presented results are interesting,
- the terminology used in the article is correct and consistent with the current norms,
- the visuals have been properly selected,
- the choice of the used literature is relevant and sufficient. In the text of the article, literature was cited into numerical terms. The references contains a list of literature in the order of quoting. The structure of the article is correct. The relevant sections are chosen in a suitable way.

Review form: Reviewer 2

Is the manuscript scientifically sound in its present form?

Yes

Are the interpretations and conclusions justified by the results?

Yes

Is the language acceptable?

Yes

Is it clear how to access all supporting data?

Yes

Do you have any ethical concerns with this paper?

No

Have you any concerns about statistical analyses in this paper?

No

Recommendation?

Accept with minor revision (please list in comments)

Comments to the Author(s)

The continuous and discontinuous surface deformations can damage buildings and infrastructure, especially the discontinuous deformation. This paper provides a good research in terms of bed separation backfill to reduce surface cracking due to mining under thick and hard conglomerate. The achievements are important for understanding the discontinuous deformation

mechanism of surface under thick and hard conglomerate. The work is potential to publish, however, there are several issues that the authors have to take care of them:

- (1) The paper needs to be revised to improve its writing style and English language. There are some wrong words, grammar, etc.
- (2) Since this journal is an international one, suggest to review and add some international references.
- (3) The authors should illustrate how to obtain the data in Table 1.
- (4) In Fig. 7, the authors should definitely mark the position of the bed separation. The author should provide the clear photos in Fig. 6.
- (5) What is the level of Luli bridge's defense? What is the maximum allowable surface deformation? Did the surface subsidence of 251 mm exceed the maximum allowable deformation?

Review form: Reviewer 3 (Grzegorz Strozik)

Is the manuscript scientifically sound in its present form?

Yes

Are the interpretations and conclusions justified by the results?

Yes

Is the language acceptable?

Yes

Is it clear how to access all supporting data?

Not Applicable

Do you have any ethical concerns with this paper?

No

Have you any concerns about statistical analyses in this paper?

No

Recommendation?

Accept as is

Comments to the Author(s)

Your paper should initiate vivid discussion among the specialists regarding the method for prediction of place and time of occurrence of bed separation zones above a coal seam during its mining extraction. In experience of Polish and Australian mining occurrence of bed separation zones is hardly predictable, because in specific point (bore hole) such a void could exist only for very limited time, while the progress of subsidence is fast (and is connected with advance of the longwall, depth, and mechanical properties of the strata in the rock mass). In Table 1 you present data, which suggest that your model simulates only densities of particular rock types.

Considering presence of gypsum and calcium carbonate you also simulate Young modulus, compressive strength and maybe other mechanical properties, what seems to be necessary, but you did not mention that aspect.

Considering large amount of fill mixture and its good flowability, you have backfilled not the bed separation zones (or not only them), but all cracks and voids in the rock mass (especially below the horizon of expected bed separation zones), what resulted obviously in limitation of

subsidence. In simple words - the reality is not so simple as one take an impression after reading your work, but nevertheless it is very valuable and interesting input into the problem of limitation of subsidence using filling of voids occurring at a depth shallower than the mining operations level.

Decision letter (RSOS-190880.R0)

28-Jun-2019

Dear Dr Yin,

The editors assigned to your paper ("Bed separation backfill to reduce surface cracking due to mining under thick and hard conglomerate - a case study") have now received comments from reviewers. We would like you to revise your paper in accordance with the referee and Associate Editor suggestions which can be found below (not including confidential reports to the Editor). Please note this decision does not guarantee eventual acceptance.

Please submit a copy of your revised paper before 21-Jul-2019. Please note that the revision deadline will expire at 00.00am on this date. If we do not hear from you within this time then it will be assumed that the paper has been withdrawn. In exceptional circumstances, extensions may be possible if agreed with the Editorial Office in advance. We do not allow multiple rounds of revision so we urge you to make every effort to fully address all of the comments at this stage. If deemed necessary by the Editors, your manuscript will be sent back to one or more of the original reviewers for assessment. If the original reviewers are not available, we may invite new reviewers.

- Data accessibility

It is a condition of publication that all supporting data are made available either as supplementary information or preferably in a suitable permanent repository. The data

accessibility section should state where the article's supporting data can be accessed. This section should also include details, where possible of where to access other relevant research materials such as statistical tools, protocols, software etc can be accessed. If the data have been deposited in an external repository this section should list the database, accession number and link to the DOI for all data from the article that have been made publicly available. Data sets that have been deposited in an external repository and have a DOI should also be appropriately cited in the manuscript and included in the reference list.

If you wish to submit your supporting data or code to Dryad (<http://datadryad.org/>), or modify your current submission to dryad, please use the following link:
<http://datadryad.org/submit?journalID=RSOS&manu=RSOS-190880>

- **Competing interests**

- **Authors' contributions**

- **Acknowledgements**

- **Funding statement**

Kind regards,

on behalf of Professor R. Kerry Rowe (Subject Editor)
openscience@royalsociety.org

Reviewers' Comments to Author:

Reviewer: 1

The article "Bed separation backfill to reduce surface cracking due to mining under thick and hard conglomerate - a case study" is written in a clear and understandable way.

In this paper, the discontinuous surface deformation mechanism in the Huafeng coal mine was analyzed through a similar material simulation test. Based on these, the bed separation backfill was proposed and adopted to control surface deformation and to protect the Luli bridge. The mechanism of surface cracking controlled by bed separation backfill was discussed.

Formal remarks:

- the summary reflects the essence of the study,
 - the presented results are interesting,
 - the terminology used in the article is correct and consistent with the current norms,
 - the visuals have been properly selected,
 - the choice of the used literature is relevant and sufficient. In the text of the article, literature was cited into numerical terms. The references contains a list of literature in the order of quoting.
- The structure of the article is correct. The relevant sections are chosen in a suitable way.

Reviewer: 2

The continuous and discontinuous surface deformations can damage buildings and infrastructure, especially the discontinuous deformation. This paper provides a good research in terms of bed separation backfill to reduce surface cracking due to mining under thick and hard conglomerate. The achievements are important for understanding the discontinuous deformation mechanism of surface under thick and hard conglomerate. The work is potential to publish, however, there are several issues that the authors have to take care of them:

- (1)The paper needs to be revised to improve its writing style and English language. There are some wrong words, grammar, etc.
- (2)Since this journal is an international one, suggest to review and add some international references.
- (3)The authors should illustrate how to obtain the data in Table 1.
- (4)In Fig.7, the authors should definitely mark the position of the bed separation. The author should provide the clear photos in Fig. 6.
- (5) What is the level of Luli bridge's defense? What is the maximum allowable surface deformation? Did the surface subsidence of 251 mm exceed the maximum allowable deformation?

Reviewer: 3

Your paper should initiate vivid discussion among the specialists regarding the method for prediction of place and time of occurrence of bed separation zones above a coal seam during its mining extraction. In experience of Polish and Australian mining occurrence of bed separation zones is hardly predictable, because in specific point (bore hole) such a void could exist only for very limited time, while the progress of subsidence is fast (and is connected with advance of the longwall, depth, and mechanical properties of the strata in the rock mass). In Table 1 you present data, which suggest that your model simulates only densities of particular rock types.

Considering presence of gypsum and calcium carbonate you also simulate Young modulus,

compressive strength and maybe other mechanical properties, what seems to be necessary, but you did not mention that aspect.

Considering large amount of fill mixture and its good flowability, you have backfilled not the bed separation zones (or not only them), but all cracks and voids in the rock mass (especially below the horizon of expected bed separation zones), what resulted obviously in limitation of subsidence. In simple words - the reality is not so simple as one takes an impression after reading your work, but nevertheless it is very valuable and interesting input into the problem of limitation of subsidence using filling of voids occurring at a depth shallower than the mining operations level.

Editorial comments to the Author:

You'll see that the reviewers are largely in favour of publication of your paper, and the Editors are in agreement, based on the referee feedback. However, as reviewer two has a number of (mostly minor) comments, and recommends the paper is sent for a language editing service (<https://royalsociety.org/journals/authors/language-polishing/>) to improve the written English, we would like you to revise the paper - this ensures you've sufficient time to complete the modifications needed, including seeking language advice, before resubmitting for consideration.

Author's Response to Decision Letter for (RSOS-190880.R0)

See Appendix A.

RSOS-190880.R1 (Revision)

Review form: Reviewer 2

Is the manuscript scientifically sound in its present form?

Yes

Are the interpretations and conclusions justified by the results?

Yes

Is the language acceptable?

Yes

Do you have any ethical concerns with this paper?

No

Have you any concerns about statistical analyses in this paper?

No

Recommendation?

Accept as is

Comments to the Author(s)

This edition of the manuscript has been revised in accordance with the reviewer's opinion, recommended to accept and publish.

Decision letter (RSOS-190880.R1)

29-Jul-2019

Dear Dr Yin,

I am pleased to inform you that your manuscript entitled "Bed separation backfill to reduce surface cracking due to mining under thick and hard conglomerate - a case study" is now accepted for publication in Royal Society Open Science.

on behalf of R. Kerry Rowe (Subject Editor)
openscience@royalsociety.org

Associate Editor Comments to Author:

Thank you for this revision - the reviewer consulted was the most critical in the earlier submission, and they are now happy for the paper to be accepted. Congratulations!

Reviewer comments to Author:
Reviewer: 2

Comments to the Author(s)

This edition of the manuscript has been revised in accordance with the reviewer's opinion, recommended to accept and publish.

Appendix A

Responses to Comments

Manuscript Number: RSOS-190880

Title: Bed separation backfill to reduce surface cracking due to mining under thick and hard conglomerate - a case study

Dear Editors and Reviewers:

We would like to re-submit the attached manuscript entitled “Bed separation backfill to reduce surface cracking due to mining under thick and hard conglomerate - a case study” for publication in the Royal Society Open Science. The manuscript ID is RSOS-190880. The manuscript has been revised based on the comments from the reviewers. The revised sections are marked in red in the paper. The responses to the individual comments of the reviewers are attached below. All language and formatting issues have also been addressed in the revised manuscript.

We thank you and the reviewers for your valuable comments, which have helped us improve the manuscript considerably. We hope that the revised manuscript will be suitable for publication.

Thank you for your consideration. I look forward to hearing from you.

Sincerely,

Dawei Yin^{1,2}, Shaojie Chen¹, Bo Li¹, Weijia Guo¹

1.State Key Laboratory of Mine Disaster Prevention and Control, Shandong University of Science and Technology, Qingdao 266590, China

2. Shandong Key Laboratory of Civil Engineering Disaster Prevention and Mitigation, Shandong University of Science and Technology, Qingdao 266590, China

Phone number: +86 532 86057948

E-mail address: csjwyb@163.com (Shaojie Chen); 1114654624@qq.com (Bo Li)

Reviewer #1:

We thank you for your comments. Our revisions based on your comments are listed below.

Comment 1: The article "Bed separation backfill to reduce surface cracking due to mining under thick and hard conglomerate - a case study" is written in a clear and understandable way. In this paper, the discontinuous surface deformation mechanism in the Huafeng coal mine was analyzed through a similar material simulation test. Based on these, the bed separation backfill was proposed and adopted to control surface deformation and to protect the Luli bridge. The mechanism of surface cracking controlled by bed separation backfill was discussed. Formal remarks:

- the summary reflects the essence of the study,
- the presented results are interesting,
- the terminology used in the article is correct and consistent with the current norms,
- the visuals have been properly selected,
- the choice of the used literature is relevant and sufficient. In the text of the article, literature was cited into numerical terms. The references contains a list of literature in the order of quoting.

The structure of the article is correct. The relevant sections are chosen in a suitable way.

Response: Thank you very much for your recognition and evaluation of this article. The manuscript has been revised based on the comments from other reviewers. All language and formatting issues have also been addressed in the revised manuscript.

Reviewer #2:

We thank you for your comments. Our revisions based on your comments are listed below.

Comment 1: The paper needs to be revised to improve its writing style and English language. There are some wrong words, grammar, etc.

Response: Thank you very much. We apologize for poor English in the initial submitted paper. We have re-edited the English in the manuscript for grammar and spelling issues, and the paper has been extensively edited by professional English editors from Mogo Editr. We hope that the quality of English now meets the standards expected from *Royal Society Open Science*.

Comment 2: Since this journal is an international one, suggest to review and add some international references.

Response: Thank you very much. We have added some international references, as follows.

1. Yin DW, Chen SJ, Liu XQ, Ma HF. 2018 Effect of joint angle in coal on failure mechanical behavior of roof rock-coal combined body. *Q. J. Eng. Geol. Hydroge.* **51**, 202-209. (doi: 10.1144/qjegh2017-041)
2. Ghasemi E, Shahriar K. 2012 A new coal pillars design method in order to enhance safety of the retreat mining in room and pillar mines. *Safety Sci.* **50**, 579-585. (doi: 10.1016/j.ssci.2011.11.005)
3. Isiaka AI, Durrheim RJ, Manzi MSD. 2018 High-resolution seismic reflection investigation of subsidence and sinkholes at an abandoned coal mine site in South Africa. *Pure Appl. Geophys.*, **176**, 1531-1548. (doi: 10.1007/s00024-018-2026-3)
4. Chen SJ, Wang HL, Wang HY, Guo WJ, Li XS. 2016 Strip coal pillar design based on estimated surface subsidence in eastern China. *Rock Mech. Rock Eng.* **49**, 3829-3838. (doi: 10.1007/s00603-016-0988-y)
5. Donnelly LJ, Reddish DJ. 1994 The development of surface steps during mining subsidence: "not due to fault reactivation". *Eng. Geol.* **36**, 243-255. (doi: 10.1016/0013-7952(94)90006-X)

Comment 3: The authors should illustrate how to obtain the data in Table 1.

Response: Thank you very much. The data in Table 1 were provided by Zhu (2011). And we have added the reference.

Comment 4: In Fig.7, the authors should definitely mark the position of the bed separation. The author should provide the clear photos in Fig. 6.

Response: Thank you very much. We have added a red wire frame to illustrate the position of the bed separation. And a clear photos in Fig. 6 were given in the revised paper.

Comment 5: What is the level of Luli bridge's defense? What is the maximum allowable surface deformation? Did the surface subsidence of 251 mm exceed the maximum allowable deformation?

Response: Thank you very much. The surface subsidence of 251 mm is located at the southern approach bridge of the Luli bridge. The maximum allowable surface subsidence of approach bridge is about 1000 mm, which is far less than that of the surface subsidence of this area.

Reviewer #3:

We thank you for your comments. Our revisions based on your comments are listed below.

Comment 1: Your paper should initiate vivid discussion among the specialists regarding the method for prediction of place and time of occurrence of bed separation zones above a coal seam during its mining extraction. In experience of Polish and Australian mining occurrence of bed separation zones is hardly predictable, because in specific point (bore hole) such a void could exist only for very limited time, while the progress of subsidence is fast (and is connected with advance of the longwall, depth, and mechanical properties of the strata in the rock mass). In Table 1 you present data, which suggest that your model simulates only densities of particular rock types. Considering presence of gypsum and calcium carbonate you also simulate Young modulus, compressive strength and maybe other mechanical properties, what seems to be necessary, but you did not mention that aspect.

Considering large amount of fill mixture and its good flowability, you have backfilled not the bed separation zones (or not only them), but all cracks and voids in the rock mass (especially below the horizon of expected bed separation zones), what resulted obviously in limitation of subsidence. In simple words - the reality is not so simple as one takes an impression after reading your work, but nevertheless it is very valuable and interesting input into the problem of limitation of subsidence using filling of voids occurring at a depth shallower than the mining operations level.

Response: Thank you very much for your recognition and evaluation of this article. The data in Table 1 were provided by Zhu (2011) for studying the mechanism of rock burst hazard in Huafeng coal mine. And we added the reference. In the simulation test on similar materials, the strength similitude between the prototype and simulation model is mainly considered. Meanwhile, the manuscript has been revised based on the comments from other reviewers. All language and formatting issues have also been addressed in the revised manuscript.